# Fabrication of Enhanced Mechanical Properties and Intrinsic Flame-Retardant Polyurethane Elastomer Containing 4-(Phenylethynyl) Di(Ethylene Glycol) Phthalate

**DOI:** 10.3390/polym13152388

**Published:** 2021-07-21

**Authors:** Meina Xie, Daikun Jia, Jin Hu, Jiyu He, Xiangmei Li, Rongjie Yang

**Affiliations:** School of Materials Science and Engineering, Beijing Institute of Technology, Beijing 100081, China; 3120185583@bit.edu.cn (M.X.); yaotiaoshunv0927@sina.com (D.J.); meowbelle@163.com (J.H.); hejiyu@bit.edu.cn (J.H.); yrj@bit.edu.cn (R.Y.)

**Keywords:** thermoplastic polyurethane elastomer, aromatic acetylene, flame-retardant, strength

## Abstract

In this study, the aromatic acetylene compound 4-(phenylethynyl) di(ethylene glycol) phthalate (PEPE) was used as a chain extender, partially replacing 1,4-butanediol. To synthesize an intrinsic flame-retardant thermoplastic polyurethane elastomer (TPU) with an aromatic acetylene structure, PEPE was synthesized by a two-step polymerization. The flame retardancy, thermal stability, and mechanical properties of TPU were studied. The microstructure of TPU char was investigated by scanning electron microscopy to analyze the flame-retardant mechanism. The tensile strength of TPU containing 1.35 wt% PEPE was 39.2 MPa, which was almost twice as much as neat TPU, showed a dramatic decrease in the peak heat release rate and total heat release, and declined by 46.2% and 24.5%, respectively. After the flame-retardant TPU burned, a cross-linked network foaming char structure was formed. The results showed that PEPE improved the mechanical properties of TPU and conferred good stability that promoted the formation of charcoal and reduced heat release during the combustion of TPU.

## 1. Introduction

Thermoplastic polyurethane elastomer (TPU) is a typical block linear polymer [1,2,3]. The rigid “hard segment” consists of a diisocyanate and a chain extender diol, while the flexible “soft segment” is composed of a polyester or polyether type oligomer diol. The “hard segment” and “soft segment” are linked via a carbamate group to form a completely linear polymer backbone with a soluble and meltable nature.

Due to differences in polarity and crystallinity, the “hard segment” and “soft segment” are thermodynamically incompatible and have a tendency to spontaneously separate into two phases. Consequently, TPU can form a microphase separation structure [4,5,6]. As result of hydrogen bonds formed by the carbamate group, the “hard segment” is glassy at room temperature and is dispersed in the rubber phase formed by “soft segments”. Subsequently, TPU molecular chains form a sea-island two-phase structure [7,8]. TPU possesses both the high strength of plastic and the high elasticity of rubber due to its unique structure, such that its performance is between a plastic and rubber material [9,10,11].

Due to its excellent characteristics, such as high strength, good elasticity, abrasion resistance and biological properties, TPU has been widely used in the fields of: aerospace, medical and health, industrial parts, textile and apparel, electrical and electronics. [12]. However, TPU is easily flammable and its combustion is accompanied by a melting, dripping phenomenon that can have serious consequences in the event of a fire [13]. The common solution is to add flame-retardants by physical blending.

Cai et al. prepared an operable platform for the functionalization of chemically inert boron nitride nanosheets for flame retardancy and toxic gas suppression of TPU. The resultant h-BN nanohybrids enhanced the flame retardancy of thermoplastic polyurethane (TPU), which was confirmed by significant reductions in peak heat release rate (23.5%) and total heat release (22.1%). The smoke product rate and total smoke release of TPU composite containing 2.0 wt% h-BN nanohybrids were decreased by 29.2% and 8.6%, respectively [14]. Shan X.Y. et al. prepared a nickel phosphate salt, NaNiP, using a hydrothermal method, and added it to TPU containing intumescent flame-retardants (IFR). The NaNiP delayed evaporation of the thermal decomposition product in the TPU/IFR system, resulting in the formation of an expanded and dense char layer, therefore increasing the residual char content [15]. Chen YH et al. designed and synthesized a new flame-retardant (IFR) containing three types of flame-retardant element, including phosphorus, nitrogen, and sulfur. Their effects on the properties of TPU composites were studied by thermogravimetric analysis (TG), scanning electron microscopy (SEM), and Fourier transform infrared spectroscopy (FTIR). It was found that sulfur and phosphorus remained in the residual carbon and helped to form a continuous dense layer of carbon, hindering heat transfer and combustible gas diffusion [16]. Liu LB et al., inspired by the hydrophobic surface structure of biomaterials, assembled the hydrophobic charring agent N-methyl triazine-piperazine copolymer (MTPC) and modified magnesium oxide (MgO) on the surface of ammonium polyphosphate (APP) to prepare a hydrophobic intumescent flame-retardant (HIFR) system. The constructed HIFR system exhibited excellent hydrophobicity, with a water contact angle of 139°. The tensile strength and elongation at the break for TPU/10 wt% HIFR composites increased by 48% and 106% compared with the values for TPU/10 wt% APP. Specimens of TPU/10 wt% HIFR composites before and after water-resistance tests achieved a UL-94 V-0 rating without dripping along with limiting oxygen index (LOI) values of 27.5% and 27.3% [17]. Additive flame-retardants, however, present disadvantages such as poor compatibility, loss of the additive, and the needed addition of larger quantities. 

In contrast, reactive flame-retardants are chemically bonded to the molecular backbone of the polymer material, which has little impact on the original properties, provides good compatibility, and confers permanent flame-retardancy. El Khatib W et al. synthesized phosphonic acid glycol using propyl or vinyl alkyl phosphonic acid and 3-mercapto-1,2-propanediol, which was used as a chain extender and flame-retardant, thus increasing the temperature stability of polyurethane elastomer to 250 °C. The residue content at 600 °C increased as the phosphorus content was increased. When the phosphorus content was greater than 0.5%, the limiting oxygen index (LOI) value was higher than 21% [18]. Wazarkar K et al. synthesized a phosphorus-reactive flame-retardant using phosphorus oxychloride and N-methylaminoethanol as raw materials and successfully introduced it into the polyurethane main chain to improve flame retardancy [19]. Tang QH et al. synthesized inherently flame-retardant and anti-dripping thermoplastic poly(imide-urethane)s (TPIU). The results indicated that the total heat release (THR) and peak heat release rate (p-HRR) of TPIU were 14.62% and 64.02% lower than the respective parameters of TPU [20]. The carbon-carbon triple bond (C≡C) of the arylalkynyl group can react at high temperatures to form a highly cross-linked and more stable network structure between molecules, significantly enhancing the charcoal properties of the polymer [21,22,23,24]. Zhao HB et al. copolymerized arylalkynyl diol with polyester (PET) and cross-linked the carbon-carbon triple bond into a network structure at a high temperature, which significantly improved the anti-dripping performance and LOI of the copolymer [25].

To date, flame-retardant research on TPU mainly incorporate flame-retardants, containing flame-retardant elements, for TPU in the form of physical blending to improve flame-retardant performance, and synthesizing phosphorus-containing reactive flame-retardants to participate in a TPU molecules chain for flame-retardant research. The thermal stability and good char-forming properties of the arylalkynyl structure have been confirmed in previous research results [26]. In this work, the aromatic arylalkynyl structure was introduced into the TPU molecular chain and the thermal stability and flame retardancy of the synthesized TPU were studied.

## 2. Materials and Methods

### 2.1. Materials

The 4-(phenylethynyl) di(ethylene glycol) phthalate (PEPE) was purchased from BIT Flame Retardant Technology Co., Ltd. (Beijing, China). Figure 1 shows the structure of PEPE. *N*,*N*-Dimethylformamide (DMF, AR) was supplied by Beijing Tongguang Fine Chemicals Company (Beijing, China). 1,4-Butanediol (BDO, AR) was purchased from Tianjin Fuchen Chemicals Reagent Factory (Tianjin, China). Diphenylmethane diisocyanate (MDI, TP) was supplied by Mitsui Chemicals Inc. (Tokyo, Japan). Polytetramethylene ether glycol (PTMG, Mn = 2000) was provided by Shanghai Aladdin Biochemical Technology Co., Ltd. (Shanghai, China).

### 2.2. Preparation of Flame-Retardant TPU

According to Scheme 1, the polymers were prepared by a two-step polymerization using MDI and PTMG, with BDO and PEPE as the chain extender. The synthesis was conducted by mechanical agitation in a three-necked flask. The appropriate amount of PTMG was poured into the flask containing melted MDI, and the reaction was stirred at 80 °C for 1 h. Suitable quantities of BDO and PEPE (as the chain extender) were dissolved in DMF and then poured into the flask. The system was stirred vigorously for an additional 5 h under a nitrogen atmosphere. Ultimately, the product was poured into a poly(tetrafluoroethylene) (PTFE) mold and then placed in a blast oven at 40 °C (to prevent formation of bubbles in the sample) for 48 h to evaporate some of the DMF. The oven was then heated at 70 °C until the sample reached constant weight. The molar ratios of reagents for various TPU/PEPE polymers are shown in Table 1, which shows the weight percent of PEPE in the total mass for each sample.

### 2.3. Measurements

#### 2.3.1. Molecular Weight of PEPE

Molecular weight was measured using a Waters Breeze TM2 HPLC Gel Permeation Chromatograph (as displayed in Appendix A for the equipment.) Monodisperse polystyrene was the standard and the solvent was tetrahydrofuran at a flow rate of 1 mL/min. The column temperature was 40 °C.

#### 2.3.2. The Mechanical Properties Test

The tensile strength and elongation at the break of the dumbbell-shaped samples were tested using a CMT4101-type microcomputer-controlled electronic universal testing machine according to the ISO 527-2 standard. The stretch rate was 100 mm/min (as observed in Appendix A for the test).

#### 2.3.3. Thermogravimetry Analysis Test

Thermogravimetric analysis (TG) of the samples under nitrogen atmosphere was performed on a NETZSCH TG 209 FI thermogravimetric analyzer (as displayed in Appendix A). The temperature range was 40–600 °C at a heating rate of 10 °C/min.

#### 2.3.4. Cone Calorimeter Test

The combustion test was performed on a cone calorimeter (R-S/FTT0007) according to ISO 5660 standard procedures, using 100 mm × 100 mm × 3 mm specimens. Each specimen was exposed horizontally to 50 kW/m^2^ external heat flux. The results were the average for three measurements (as observed in Appendix A for the test).

#### 2.3.5. Limiting Oxygen Index Test

The LOI was obtained on a Rheometric scientific limiting oxygen indexer according to ISO 4589-2 standard with 100 mm × 6 mm × 3 mm specimens (as displayed in Appendix A for the equipment).

#### 2.3.6. Scanning Electron Microscopy

The morphologies of char residues after the combustion test, coated in advance with a gold layer, were observed using a QUANTA FEG 650 thermal field scanning electron microscope (SEM).

## 3. Results and Discussion

### 3.1. Molecular Weight and Mechanical Properties

Mechanical properties, one of the important parameters for TPU, are closely related to molecular weight. The data in Table 2 show that the synthesized TPU samples were of high molecular weight (mostly above 43,000). Due to its high rigidity, the incorporation of PEPE leads to a broader molecular weight distribution. Greater amounts of PEPE in TPU disfavor the formation of higher-molecular-weight products. Moreover, a moderate number of PEPE structures in the TPU molecular chain conferred greater tensile strength. In particular, the tensile strength of 1.35% PEPE-TPU (39.2 MPa) was twice that of Blank-TPU. When the PEPE content in the hard segment of the TPU molecular chain reaches a certain level, the microphase separation effect formed by the soft and hard segments was in appropriate proportion, resulting in stronger rigidity of the hard segment and increased flexibility of the soft segment. As the PEPE content continued to increase, the degree of branching of the TPU molecular chain and the distance between molecular chains was increased, causing the force between the molecular chains to decrease and resulting in a decrease in tensile strength. While the flexibility of the soft molecular chain was not greatly affected, an apparent flexibility-increasing trend in the elongation at the break continued to increase.

### 3.2. Thermal Stability and LOI

The thermal stability of the polymer plays an important role in determining the applications for which it can be used. The data in Table 3 (and Figure 2) shows that the temperature at which TPU decomposed, 5% (T_5%_), was very low when PEPE was used as a chain extender. The maximum weight loss rate (MLR_max_) decreased from −22.31%/min to −19.23%/min as the PEPE content increased. The residual char increased significantly from 0.27% for Blank-TPU to 6.74% for the 5.25%PEPE-TPU. In addition, the LOI increased from 19.8 for the blank sample to 21.2 for 5.25%PEPE-TPU.

The TG results showed that PEPE improved the stability and charcoal properties of TPU, which enhanced the flame-retardant effect to a certain degree. This result is consistent with previous studies on the properties of polymers with flame-retardants containing aromatic acetylene structures [19,21,23]. It was confirmed that compounds containing aromatic acetylene groups were cross-linked during the combustion of TPU to promote the formation of carbon residue, which had a certain flame-retardant effect.

### 3.3. Cone Calorimetry

In the combustion test, a cone calorimeter was used to investigate the fire hazards of TPU and its composites. Cone calorimetry can provide crucially relevant parameters, such as the peak heat release rate (p-HRR), total heat release (THR) and time to ignition (TTI) [27]. In flame-retardant TPU samples, p-HRR (Figure 3a) and THR (Figure 3b) were greatly reduced. The p-HRR of Blank-TPU decreased from 1476.4 to 794.4 kW/m^2^ for 1.35%PEPE-TPU, while THR decreased from 102 to 64.96 MJ/m^2^ for 2.68%PEPE-TPU, corresponding to 46.2% and 36.3% reductions, respectively. Furthermore, the values for TTI (Table 4) improved by approximately 2 s. From the digital photographs of char shown in Figure 4, more residue was observed in 5.25%PEPE-TPU, while the Blank-TPU was very clean. This was due to the cross-linked structure formed by PEPE at elevated temperatures, which effectively inhibited combustion.

### 3.4. Scanning Electron Microscopy

The microstructure of the TPU residual char after cone calorimetry was observed. It was found that PEPE-TPU formed a cross-linked network structure after combustion, with gaps among the cross-linked networks, as shown in Figure 5. This was consistent with the aromatic acetylene structure cross-linking system shown in Scheme 2.

## 4. Conclusions

The intrinsic flame-retardant thermoplastic polyurethane elastomers with aromatic acetylene flame-retardant PEPE were synthesized. It was found that partial replacement of BDO with PEPE as a chain extender dramatically improved the mechanical and flame retardancy properties of TPU. Due to its high rigidity, a relatively high molecular-weighted TPU can be synthesized when a proper amount of PEPE was incorporated. The tensile strength was significantly increased to 39.2 MPa for 1.35%PEPE-TPU. It was previously known that the maximum weight loss rate of TPU with a PEPE structure reduced to 19.23%/min, and the residual char increased to 6.74% for only 5.25% PEPE. Importantly, the peak heat release rate and the total heat release for 1.35% PEPE drastically reduced 46.2% and 24.5%, respectively. This result confirmed that compact cross-linked char structure was helpful to improve flame retardancy. The introduction of the arylalkynyl structure into the TPU molecular chain to synthesize intrinsically flame-retardant TPU leads to an integrated performance.

## Data Availability

The data presented in this study are available on request from the corresponding author.

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
