# Peer review of "Fabrication of Enhanced Mechanical Properties and Intrinsic Flame-Retardant Polyurethane Elastomer Containing 4-(Phenylethynyl) Di(Ethylene Glycol) Phthalate"

_polymers, 2021, doi:10.3390/polym13152388_

Round 1

Reviewer 1 Report

  1. The current study investigates the of what the authors call “Excellent!” mechanical properties and fire-resistant elastomers that contain aromatic compounds. For this, the authors carry out tests to study the microstructure and flame resistance and mechanical properties. The authors report that the PEPE improved mechanical properties and reduced heat during the burning of the elastomer.
  2. The abstract needs significant modifications. Please consider reviewing the abstract and highlight the novelty, major findings, and conclusions.
  3. Please consider either removing Figure 1 or moving it to the materials and methods section
  4. Please rename section 2 to materials and methods section
  5. After line 83 the authors should attempt to answer the following question: What is the research gap did you find from the previous researchers in your field? Mention it properly. It will improve the strength of the article.
  6. Please expand more in lines 84-85 the last paragraph in the introduction should give more details that what you stated there.
  7. Experimental change to Materials and methods
  8. The authors should add tables for representing the mechanical properties of the materials used in their study.
  9. Scheme 1. Some of the data on the right side seems to be missing.
  10. Check line 101 and 106 the Celsius degree symbol have some issue.
  11. 2.3. Measurements this section does not read well at all. Please combine all these small paragraphs into one large paragrpah
  12. The materials and methods section is weak, it lacks any images of the equipment used or tools for fabricating the samples, there is not even photos of the fabricated materials or images of the test equipment used for the fire testing..etc. This is mainly an experimental study, and the authors should give full details of their experimental work so that readers can better understand what was done here.
  13. The results are merely described and is limited to comparing the experimental observation. The authors are encouraged to include a discussion section and critically discuss the observations from this investigation with existing literature.

Author Response

Point 1: The current study investigates the of what the authors call “Excellent!” mechanical properties and fire-resistant elastomers that contain aromatic compounds. For this, the authors carry out tests to study the microstructure and flame resistance and mechanical properties. The authors report that the PEPE improved mechanical properties and reduced heat during the burning of the elastomer.

Response 1: Thank you very much for your constructive comments. According to the experimental results of the current test,only 1.35% of the PEPE structure was incorporated in the molecular chain of the TPU, which can improve the integrated performance of the TPU, the tensile strength is twice the original, and the heat release and total heat release are also significantly decreased. Excellent emphasized on the improvement of mechanical and fire-resistant properties.

Point 2: The abstract needs significant modifications. Please consider reviewing the abstract and highlight the novelty, major findings, and conclusions.

Response 2: Thank you for your comments. The abstract has been modified and emphasized the novelty of the content of the article, the main findings and conclusions. Kindly please see the red font in the revised resubmitted manuscript abstract.

Point 3: Please consider either removing Figure 1 or moving it to the materials and methods section.

Response 3: Thank you for your comments. Figure 1 has been moved to the materials and methods section. Kindly please see the red font and Figure 1 in the revised resubmitted manuscript 2.1. Materials.

Point 4: Please rename section 2 to materials and methods section.

Response 4: Thank you very much for your comments. Section 2 has been renamed as the materials and methods. Kindly please see the red font in revised resubmitted manuscript section 2.

Point 5: After line 83 the authors should attempt to answer the following question: What is the research gap did you find from the previous researchers in your field? Mention it properly. It will improve the strength of the article.

Response 5: Thank you very much for your constructive comments and suggestions. At the end of the introduction, the research gaps of previous researchers in this field have been summarized. Kindly please see the red font at the end of the introduction of the revised resubmitted manuscript.

Point 6:  Please expand more in lines 84-85 the last paragraph in the introduction should give more details that what you stated there.

Response 6: Thank you very much for your constructive comments. The purpose and performance study of introducing the arylalkynyl structure into the TPU molecular chain has been included in the last paragraph of the introduction, and the corresponding references [26] was introduced. Kindly please see the red font at the end of the introduction of the revised resubmitted manuscript.

Point 7: Experimental change to Materials and methods.

Response 7: Thank you for your comments. The Experiment has been changed to Materials and methods. Kindly please see the red font in revised resubmitted manuscript section 2.

Point 8: The authors should add tables for representing the mechanical properties of the materials used in their study.

Response 8: Thank you very much for your constructive suggestions. At present, our research on the mechanical properties of TPU mainly focuses on tensile strength and elongation at break, and hardness and tear strength will be involved in subsequent studies.

Point 9: Scheme 1. Some of the data on the right side seems to be missing.

Response 9: Thank you very much for your comments. We feel sorry for these mistakes. Scheme 1. Now it has been placed in a suitable position and the entire molecular structure can be seen completely. Kindly please see the revised resubmitted manuscript 2.2. Preparation of flame retardant TPU.

Point 10: Check line 101 and 106 the Celsius degree symbol have some issue.

Response 10: Thank you very much for your comments. We feel sorry for these mistakes. The Celsius degree symbol issue in this place has been modified. Kindly please see the red font in revised resubmitted manuscript 2.2. Preparation of flame retardant TPU.

Point 11: 2.3. Measurements this section does not read well at all. Please combine all these small paragraphs into one large paragraph.

Response 11: Thank you very much for your constructive comments. Due to consideration of the twelfth suggestions, 2.3. Measurements was readjusted, and the equipment used in the test and the image of the test process are included in the Electronic Supporting Information. Kindly please see the revised resubmitted manuscript 2.3. Measurements.

Point 12: The materials and methods section is weak, it lacks any images of the equipment used or tools for fabricating the samples, there is not even photos of the fabricated materials or images of the test equipment used for the fire testing etc. This is mainly an experimental study, and the authors should give full details of their experimental work so that readers can better understand what was done here.

Response 12: Thank you very much for your constructive comments and suggestions. The image of the equipment used and images of the test equipment used for mechanical properties testing and the fire testing in the Electronic Supporting Information. Kindly please see the red font in revised resubmitted manuscript 2.3. Measurements and Electronic Supporting Information Figure S1-S5.

Point 13: The results are merely described and is limited to comparing the experimental observation. The authors are encouraged to include a discussion section and critically discuss the observations from this investigation with existing literature.

Response 13: Thank you very much for your constructive suggestions. The discussion of the results has been included in the conclusion. Kindly please see the red font in revised resubmitted manuscript 4. Conclusion.

Reviewer 2 Report

The authors introduced an aromatic acetylene compound as chain extender to synthesize a thermoplastic polyurethane elastomer. The approach seems to be new. The authors also charatarize this material, the emphasize is put on the flame retardation. The manuscript is clearly written. Please find below several comments that have to be addressed before I can recommen the paper for publication.

1) The title should be changed. It is not possible to fabricate excellent properties but it is possible to syntheisize a polymer with enhanced properties.

2) The authors should better explain why they selected an romatic acetylene structure to be introduced into the TPU.

3) I suggest to classify this paper as a Communication because it is rather short.

4) Table 2. Why is there a local maximum in the tensile strength but montonic increase in elongation?

5) The discussion in 3.1 is too short. What is meant by improved mechanical properties?

Round 2

Reviewer 1 Report

All questions answered, paper can be accepted

Reviewer 2 Report

The authors reacted properly to my comments. The paper can be recommended for publication. However, the title has to be changed because it is not correct. I suggest:

Fabrication of intrinsically flame retardant polyurethane elastomer containing 4-(phe- 3nylethynyl) di(ethylene glycol) phthalate with enhanced mechanical properties.

It is not possible to fabricate mechanical properties.